# The Migration Law of Iron during the Process of Water Icing

**Yuanqing Tang, Yan Zhang \*, Wanli Zhao, Tongshuai Liu and Yucan Liu**

College of Civil Engineering, Yantai University, Yantai 264000, China; T1073060145@163.com (Y.T.);
anytiny@163.com (W.Z.); L13053785634@163.com (T.L.); liuyucanfendou@163.com (Y.L.)
\* Correspondence: zhangyanytu@ytu.edu.cn; Tel.: + 86-135-6259-1712

**Abstract:** In this study, we utilized simulated icing experiments to investigate the effect of icing thickness, freezing temperature and initial concentration on the migration of iron in the ice–water system during water icing. The distribution coefficient "K" (the ratio of the average concentration of iron in the ice to that in the under-ice water) was used to describe the effect. The results indicated that iron partitioned stronger to under-ice water than to ice during the process of water icing, resulting in the concentration of iron in ice–water system before and after freezing being expressed as: ice < pre-freezing water < under-ice water. K decreased with the increase in icing thickness, freezing temperature and initial concentration. The temperature change in the solution will change the solubility of the solvent, so we explained the migration of iron during the process of water icing from the perspective of solid–liquid equilibrium theory. Too high or too low iron concentration may inhibit the growth of algae, thus affecting the underwater ecological environment. We expect that our study will arouse researcher's attention to the change in iron concentration in shallow lakes and ponds at high latitudes during the icebound period.

**Keywords:** simulated icing; migration law; distribution coefficient "K"; solid–liquid equilibrium theory; iron

---

## 1. Introduction

The icebound period is an important hydrological feature of surface water bodies in high-latitude regions. Within a >150-day icebound period, >50 million lakes are frozen every year [1,2]. The snow covering the surface of the ice reduces the light penetration rate [3,4], and when the snow cover on the surface reaches 10 cm, the light penetration will be reduced to a level insufficient for photosynthesis [5]. The ice cover is an obstacle to the volatilization and diffusion of the pollutants in the under-ice water [6]. As well as this, the re-oxygenation process is hindered, which lowers the dissolved oxygen concentration [7–10], inhibiting photolysis and biodegradation [11–13]. Therefore, the efficiency of the water body's self-purification declines, and the under-ice water environment has its particularity during the icebound period [14,15].

At present, some researches mainly focus on the under-ice ecology of natural water body, indicating that under-ice ecology is dynamic, and could affect the plankton dynamics in spring and summer [16,17]. Other research focuses on the effects of the freezing process on under-ice biological properties (e.g., chlorophyll a, plankton density) [18,19], chemical properties (e.g., nitrogen, phosphorus, and dissolved organic carbon contents) [20–22], and the monitoring of physical properties (e.g., water temperature, ice thickness, freezing rate) [23,24]. There are relatively few indoor simulation studies on the icing process [25]. The few studies that exist mainly focus on the migration of organic pollutants and seawater desalination [26–31], and aim at the removal of pollutants. In addition, some studies utilize the first principle to explain the migration of various heavy metal ions (Fe, Cu, Mn, Zn, Pb, Cd,

Hg) from the perspective of energy change in the freezing process, which indicate that the ability of different kinds of heavy metals to migrate from ice to water is related to their binding energy in ice [32]. However, there are relatively few studies on the migration mechanism and influencing factors of pollutants during icing.

Iron is one of the most typical pollutants in China's surface water environment [33]. Statistics show that 331 million people live in areas of China where iron levels exceed drinking water quality standards, and high concentrations of iron may cause various diseases. Related studies have shown that the body absorbs iron more than 10–20 times and may cause chronic poisoning [34,35], even leading to coma, convulsion, respiratory failure, cardiac arrest or cancer [36,37]. Where the iron content in water increases significantly due to icing, it may harm aquatic organisms and even human health [33]. Therefore, it is of necessity to clarify the migration law and mechanism of iron in the water icing process. In our research, the distribution of iron in ice and under-ice water during the water icing process was studied through simulated icing experiments. We explained the migration mechanism of iron during water icing using solid–liquid equilibrium theory. In addition, we investigated the effects of freezing temperature, icing thickness and initial concentration on iron migration.

## 2. Materials and Methods

### 2.1. Icing Simulation Device

Our study utilized an open unidirectional downward icing simulator in order to simulate the top-down icing process of natural water bodies (Figure 1). A glass cylindrical barrel (diameter 20 cm, height 37.5 cm, and volume 11,775 cm$^3$) was wrapped with Expanded Polystyrene (EPS) insulation to block the transfer of heat between the barrel and the outdoors. A temperature-controlled heating sheet was placed between the outer part of the barrel and the thermal insulation to facilitate the removal of the ice sample. A resistance wire (nickel–chromium alloy) was placed in the barrel to measure ice thickness. The distance between node 'a' and node 'b' was measured in advance as L1. A 12 V battery was used to electrify the resistance wire for 3 s. The resistance wire was raised to node 'b' to contact the ice bottom, and the distance between 'a' and the ice surface was measured as L2, L1-L2 is ice thickness. This device was then placed in a low-temperature testing box. During the experiment, the temperature deviation from the target temperature did not exceed ±0.5 °C, which is caused by the operation of the refrigerator compressor.

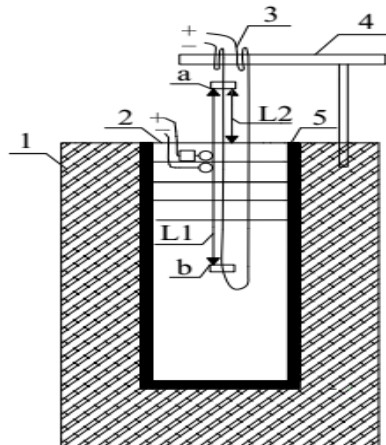

**Figure 1.** Icing simulation device: (1) Expanded polystyrene (EPS) insulation; (2) Heating sheet; (3) Device for measuring ice thickness; (4) T-shaped bracket; (5) Glass cylindrical barrel.

*2.2. Experimental Design*

(1)　To study the effect of icing thickness on the migration law of iron during water icing, we prepared the iron standard solution with the concentration of 0.6 mg/L in five icing simulation devices. Subsequently, we placed 8 L water sample (about 4/5 of the volume of the device) in each icing simulation device and placed the devices in the low-temperature testing box at −15 °C. The samples were removed when the thickness of ice reached 4 cm, 8 cm, 12 cm, 16 cm and 20 cm;

(2)　To study the effect of freezing temperature on the migration law of iron during water icing, we prepared the iron standard solution with a concentration of 0.6 mg/L in five icing simulation devices. Subsequently, we placed 8 L water sample in each icing simulation device and placed the devices in the low-temperature testing box at −5 °C, −10 °C, −15 °C, −20 °C and −25 °C. The samples were removed when the thickness of the ice reached 12 cm;

(3)　To study the effect of initial concentration on the migration law of iron during water icing, according to 'Chinese standards for drinking water quality' [38], wherein the concentration of iron in drinking water should not exceed 0.3 mg/L, the iron standard solution concentrations were set to 0.3 mg/L, 0.6 mg/L, 0.9 mg/L, 1.2 mg/L, and 1.5 mg/L. Iron standard solutions with the above concentrations were respectively placed in five icing simulators. A total of 8 L of water sample was put in each simulator device placed in the low-temperature testing box at −15 °C. The samples were removed when the thickness of the ice reached 12 cm.

The ice samples that were obtained from the simulated icing experiments were cut per 4 cm with a cutting device heated by a resistance wire (nickel–chromium alloy). Then, the ice samples were divided into several sections from the top to the bottom: 0–4 cm, 4–8 cm, 8–12 cm, 12–16 cm and 16–20 cm, which melted in a beaker at room temperature. At the same time, the under-ice water was evenly mixed and then removed and placed in beakers.

*2.3. Detection Method*

Iron standard stock solution (100 mg/L): accurately weigh 0.7020 g of ferrous ammonium sulfate $((NH_4)_2Fe(SO_4)_2 \cdot 6H_2O)$, dissolve in 50 mL of sulfuric acid (1:1), transfer to a 1000 mL volumetric flask, and add water to the standard Line, shake well. The iron standard solution in the simulated freezing experiment was obtained by diluting the iron standard stock solution (100 mg/L) with fresh deionized water (conductivity less than 0.3 μs/cm).

In this study, the total iron content in water samples and the ice-melt water samples was measured. The water samples were not filtered, and 50 mL of well-mixed water samples were placed in a 150 mL conical flask. We added 1 mL of hydrochloric acid (1:3) and 1 mL of 10% hydroxylamine hydrochloride solution. Then, we heated and boiled until the volume was reduced to about 15 mL, cooled it to room temperature, and transferred it to a 50 mL colorimetric tube. Next, we added a small piece of Congo red test paper and dropped the saturated sodium acetate solution until the test paper just turns red. We added 5 mL of buffer solution (40 g of ammonium acetate and 50 mL of glacial acetic acid diluted to 100 mL with water), added 2 mL of 0.5% phenanthroline solution, added water to the mark, and shook well. After standing for 15 min., the absorbance was measured with a 10 mm cuvette at a wavelength of 510 nm while using water as a reference. Based on the obtained absorbance and calibration curve, the total iron content was calculated in the water samples.

The content of total iron in ice water samples and ice melt water samples was determined without filtration. Following the 'Water and Wastewater monitoring analysis method' [39], each experimental product was analyzed three times and the standard deviations of these triplicates were within 5% of their arithmetic means. The experiment was carried out in the water analysis laboratory, college of civil engineering, Yantai University.

### 2.4. Data Analysis

The distribution coefficient (K) was the ratio of the average concentration of iron in the ice to that of iron in the under-ice water. It reflected the ability of iron to migrate into under-ice water in the water freezing process.

$$K = Ci/Cw \tag{1}$$

where Ci is the average concentration of iron in the ice, and Cw is the average concentration of iron in the under-ice water.

## 3. Results and Discussion

### 3.1. Migration Law of Iron in Ice–Water System during the Icing Process

The distribution of iron in the ice–water system before and after water freezing was expressed as: ice body < pre-freezing water body < under-ice water body (Figure 2). The results indicated that iron partitioned stronger to under-ice water than to ice during the process of water icing. We explained the migration of iron during the process of water icing from the perspective of solid–liquid equilibrium theory.

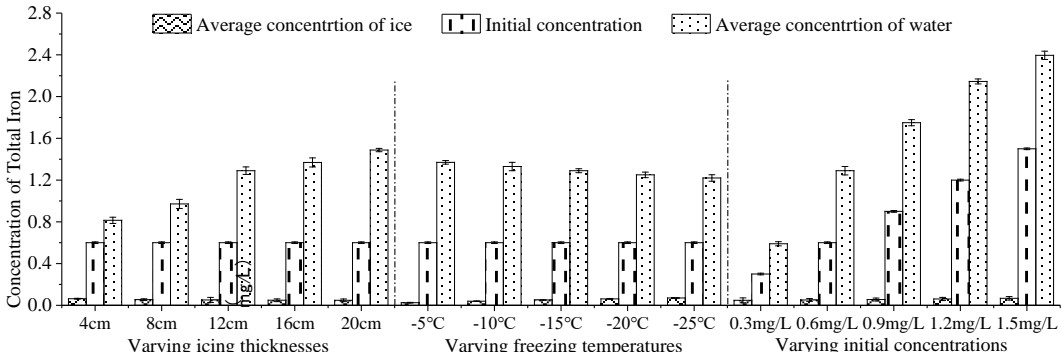

**Figure 2.** Distribution of iron in ice–water system during water icing. (Error bar: Mean ± S.D., K: the ratio of the average concentration of iron in the ice to that in the under-ice water).

Usually, the temperature–concentration phase equilibrium diagram of the solution is drawn by the solubility method (Figure 3). The T-C diagram is composed of temperature as the ordinate and concentration as the abscissa. As shown in the phase diagram, there are three curves. The DE line is the freezing point curve of $H_2O$. It can be seen from the figure that the increase in the concentration of iron makes the freezing point of $H_2O$ continuously drop. The EF line is the saturation solubility curve of iron in water, which ends at the boiling temperature of the solution and cannot be arbitrarily extended. $T_E EE_0$ is a three-phase equilibrium coexistence line of $H_2O(s)$, iron(s) and iron solution (l). There are four phase zones. Above the DEF curve is the single-phase zone (L1) of iron-unsaturated solution (l). The range of the $DET_E$ curve is the two-phase equilibrium region (A + L2) of $H_2O(s)$ and iron solution (l). To the right of the EF line and above the $EE_0$ line are two-phase equilibrium regions (B + L3) of iron (s) and iron-saturated solution (l). Below the $T_E EE_0$ line is the two-phase coexistence zone (A + B) of $H_2O(s)$ and iron(s). The iron-unsaturated solution begins to cool down from the initial temperature $T_0$. When the temperature drops to $T_1$, the concentration of the solution will increase from $C_0$ to $C_1$. At this point, the solution begins to freeze. As the temperature continuously drops to $T_2$, the ice crystals continue to form, grow and precipitate in the solution. Meanwhile, it becomes the mixture of $H_2O(s)$ and iron solution. Point E is the low eutectic point of the solid–liquid equilibrium system, and eutectic transformation will occur when the temperature drops below $T_E$. For example, when the temperature drops to $T_3$, $H_2O(s)$ and iron(s) will be precipitated out of the iron solution at the same time [40].

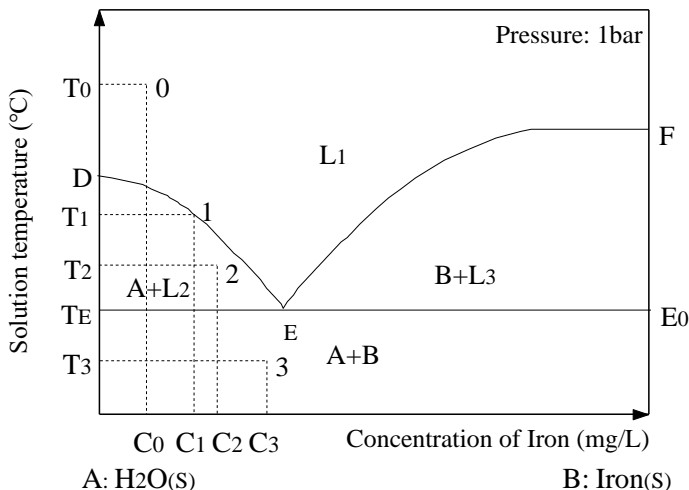

**Figure 3.** A schematic diagram of the solution's temperature–concentration phase equilibrium curve. (L1: phase region of iron-unsaturated solution; A + L2: two-phase equilibrium region of H$_2$O(s) and iron solution(l); B + L3: two-phase equilibrium regions of iron (s) and iron-saturated solution (l); A + B: two-phase coexistence zone of solid H$_2$O (H$_2$O(s)) and solid iron (iron(s))).

Ice is composed of a crystal arrangement of water molecules, and the structure of the ice crystal depends on the interaction force between its molecules. The interaction force is determined by the electronic structure of these water molecules [41]. The electrostatic attraction between the charges is very obvious, resulting in a strong interaction between the molecules, which also gives the ice a strong mechanical strength [42]. Near the solid–liquid interface, the water molecules were precipitated by hydrogen bonding and adhered to the bottom of the ice. At the same time, iron was squeezed out and escaped to the water body under the ice [43]. That is, iron migrated from the ice body to the water body in the process of water icing. Therefore, the concentration of iron in the ice body was lower than that in the under-ice water body.

However, the inside of ice still contains a certain amount of pollutants, which can be explained by the growth process of the ice body. Firstly, the direct contact between the water body and the cold source during the initial icing period resulted in a rapid icing rate, which caused some pollutants to be trapped in the upper ice [44]. After the formation of surface ice bodies, the newborn ice bodies can only grow vertically downwards due to the limitation of the surrounding ice crystals (Figure 4). Most of the ice crystals grow in a dendritic structure, so some pollutants are captured in the ice body [45].

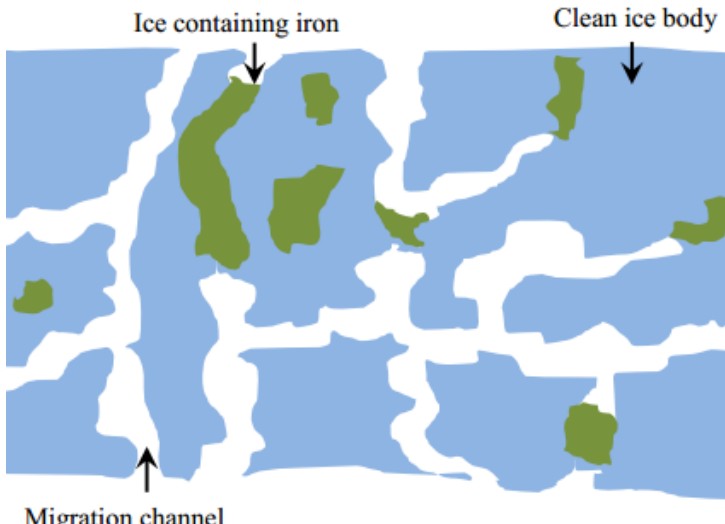

**Figure 4.** Schematic diagram of ice body profile structure [14].

### 3.2. Effects of Ice Thickness on the Migration of Iron

In the five experiments (Figure 5), iron captured in the ice body continuously moved to the lower ice body through the pore channel at the beginning of freezing [46], so the concentration of iron in the upper ice layer decreased with the increase in ice thickness, such as the concentration of iron: 4 cm ice body (0–4 cm) > 8 cm ice body (0–4 cm) > 12 cm ice body (0–4 cm) > 16 cm ice body (0–4 cm) > 20 cm ice body (0–4 cm). When the ice thickness was 12 cm, 16 cm and 20 cm, the iron concentration in the continuous ice layer decreased at first and then increased, because the ice body formed a dendritic structure pollutant-transport channel in the freezing process (Figure 4). At the beginning of icing, the icing rate is faster, increasing the pore channel density and iron content in the upper ice body. In the middle phase of icing, as the thickness of the ice increased, and the heat exchange between the under-ice water and the outside was weakened. The rate of growth of ice was reduced [47], resulting in the decrease in the amount of iron trapped in the newly formed ice body, so the concentration of iron was relatively low in middle ice bodies. In the later phase, with the obvious increase in iron concentration in the under-ice water, the concentration of iron captured in the ice cell and pore channel per unit volume were very high. In addition, iron captured in the ice body continuously migrates to the lower ice body through the pore channel, resulting in a large concentration of iron in the lower ice body. Therefore, when the ice thickness is 12 cm, 16 cm and 20 cm, the distribution of iron concentration in the continuous ice decreases first and then increases.

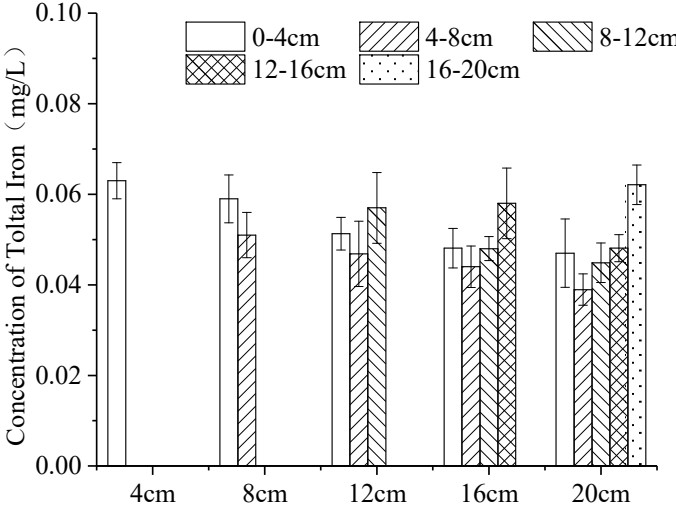

**Figure 5.** Distribution of iron at varying icing thicknesses in the ice body. (Error bar: Mean ± S.D., K: the ratio of the average concentration of iron in the ice to that in the under-ice water).

When the ice thickness was 4 cm, 8 cm, 12 cm, 16 cm and 20 cm, K was 0.077, 0.057, 0.040, 0.036 and 0.032 (Figure 6), respectively. With the increase in ice thickness, the K decreased, and the ability of iron to migrate into the under-ice water increased. This was because, in the early phase of icing, iron trapped in the ice body continued to migrate to the under-ice water through the pore channels, [46]. In the middle phase of icing, the newly-formed ice crystals became larger and the ability to capture iron decreased. There were also fewer channels for discharging iron in the ice-per-unit area, which were converted into a series of iron cells and air cells [48]. The average concentration of iron in the ice body gradually decreased as the thickness of the ice increased. In the three experiments, K reached a minimum when the ice thickness was 20 cm. Because the experimental device is relatively small, the iron concentrations in under-ice water might also be affected by the volumetric ratio of ice to water, and not only by the absolute ice thickness. Our study should thus be most relevant for shallow lakes and small ponds, with relatively thick ice cover compared to the under-ice water.

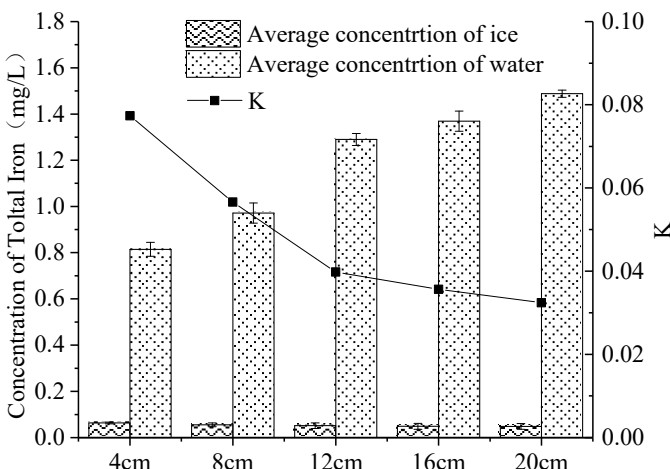

**Figure 6.** Distribution of iron of varying icing thicknesses in the ice–water system. (Error bar: Mean ± S.D., K: the ratio of the average concentration of iron in the ice to that in the under-ice water).

### 3.3. Effects of Freezing Temperatures on the Migration of Iron

The distribution of iron in continuous ice bodies also showed a tendency to decrease first and then increase (Figure 7). When the temperature was −5 °C, −10 °C, −15 °C, −20 °C and −25 °C, K was 0.019, 0.030, 0.040, 0.050 and 0.058 (Figure 8), respectively. K increased with the decrease in temperature, which means that the lower the freezing temperature, the weaker the ability of iron to migrate into the under-ice water, and the more iron was trapped in the ice.

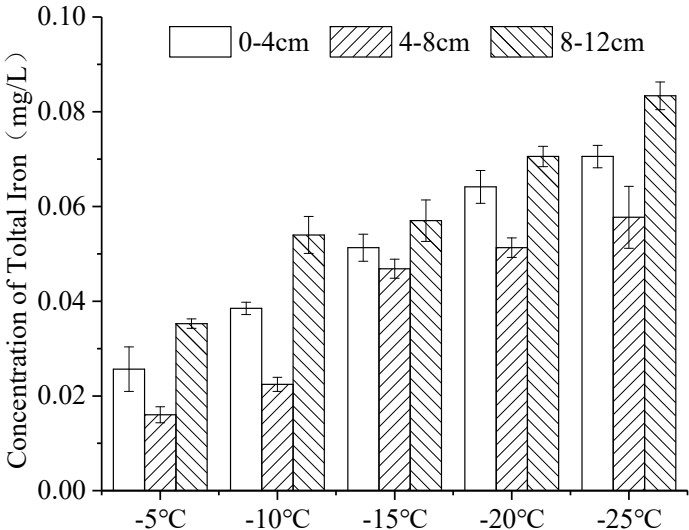

**Figure 7.** Distribution of iron at varying freezing temperatures in the ice body. (Error bar: Mean ± S.D., K: the ratio of the average concentration of iron in the ice to that in the under-ice water).

In fact, the direct factor affecting the K is the growth rate of ice, which increases with the decrease in freezing temperature. As the growth rate of ice increases, the ice crystals formed finer and denser branches, capturing more iron in the ice body, and the water molecules, meanwhile, moved faster toward the solid–liquid interface. Once that speed exceeded the speed at which iron moved toward the interface, iron was trapped by the ice crystals, [49]. Schmidt et al. also showed that the concentration of pollutants (organic nitrogen, organic phosphorus, etc.) was higher in the under-ice water of the lake when the icing was slow [50,51]. Waller and Terwilliger [52,53] studied the effect of the freezing rate

on the discharge of salt from the ice body, and they found that lower icing temperatures resulted in more saltwater inclusions in the ice.

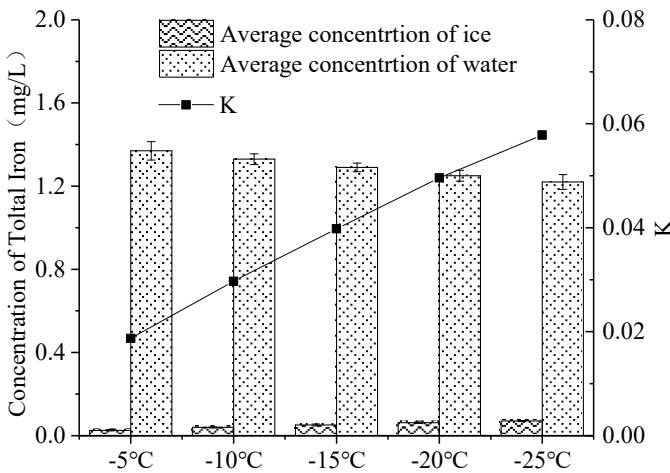

**Figure 8.** Distribution of iron at varying freezing temperatures in the ice–water system. (Error bar: Mean ± S.D., K: the ratio of the average concentration of iron in the ice to that in the under-ice water).

*3.4. Effects of Initial Concentrations on the Migration of Iron*

The distribution of iron in continuous ice bodies also showed a tendency to decrease first and then increase (Figure 9). When the initial concentration was 0.3 mg L, 0.6 mg/L, 0.9 mg/L, 1.2 mg/L and 1.5 mg/L, K was 0.081, 0.040, 0.032, 0.028 and 0.027 (Figure 10), respectively. That is, the greater the initial concentration of iron was in the non-freezing period, the greater the concentration of iron was in the ice after icing (Figure 9), and the decrease in K with the initial concentration. This was consistent with the experimental results of W. Gao [54].

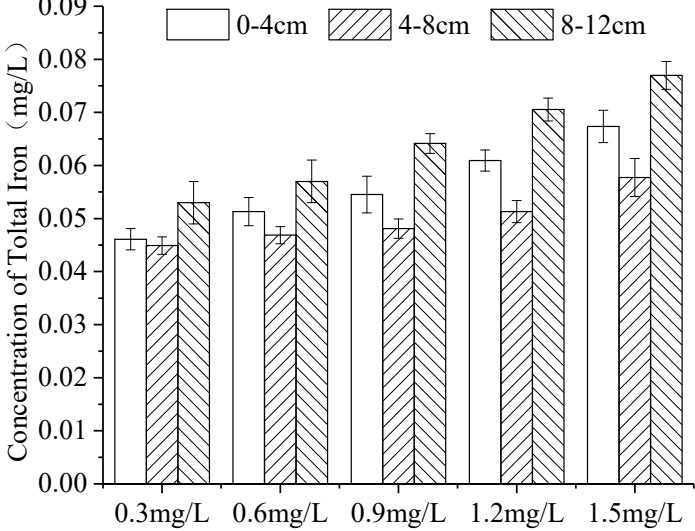

**Figure 9.** Distribution of iron with varying initial concentrations in the ice body. (Error bar: Mean ± S.D., K: the ratio of the average concentration of iron in the ice to that in the under-ice water).

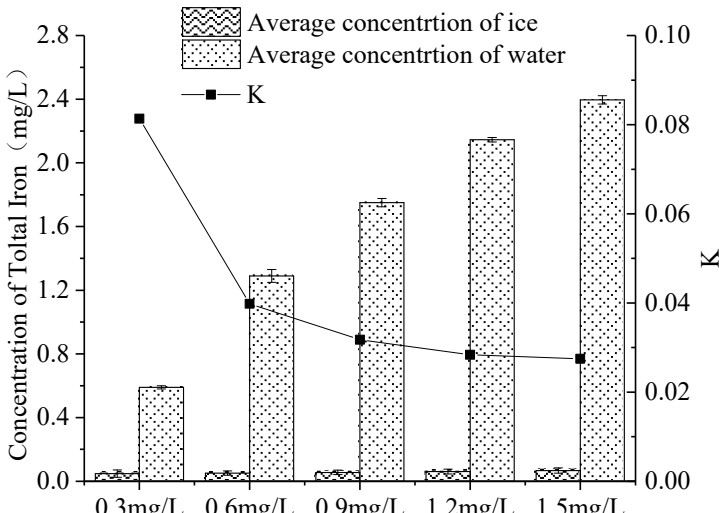

**Figure 10.** Distribution of iron with varying initial concentrations in the ice-water system. (Error bar: Mean ± S.D., K: the ratio of the average concentration of iron in the ice to that in the under-ice water).

As the initial concentration (Figure 10) of iron increased, the viscosity increased, and the diffusion decreased. The probability of a collision of ice crystals increased, and the iron was more likely to be trapped when the ice grew, so that more iron remained in the ice [55,56]. As the initial concentration of iron increased (Figure 10), more iron was trapped in the ice during the initial icing period. However, with the increase in ice thickness, the total amount of iron that migrated from the ice to the under-ice water increased. The increase in iron in the ice is much smaller than that in the under-ice water, which resulted in the decrease in K.

## 4. Conclusions

Based on the results of the natural icing simulations, the following conclusions were drawn:

(1) During icing, iron partitions more strongly to under-ice water than to ice, which may deteriorate the under-ice water environment in shallow lakes in high latitudes. In addition, iron affects the growth of phytoplankton. Too high or too low iron concentrations may inhibit the growth of algae. Thus, iron pollution caused by the icebound period should be further studied in the future;

(2) In the simulated icing experiment, the concentration of iron in the ice body showed a tendency to decrease first and then increase. K (the ratio of the average concentration of iron in the ice to that in the under-ice water) decreased with the increase in icing temperature, icing thickness, and initial concentration. That is, a higher icing temperature, larger icing thickness, and higher initial concentration are favorable for the migration of iron to under-ice water bodies. For shallow lakes, the volume ratio of ice water may be an important factor which causes the increase in iron concentration in under-ice water, too. It is hoped that this study can provide a theoretical basis for the change in iron concentration in the icing process of surface water, and provide data support for the treatment of iron-containing wastewater by using the freezing concentration effect;

(3) The distribution of iron in natural waters was affected by various factors, such as organic ligands, pH, and dissolved oxygen. In the simulated icing experiment, this study only considered the effects of icing thickness, freezing temperature and initial concentration on the migration distribution of iron. Other factors that may affect the migration of iron remain to be further explored.

**Author Contributions:** Data curation, Y.T., Y.Z., W.Z. and Y.L.; Formal analysis, Y.Z., Y.T. and T.L.; Methodology, Y.Z., Y.T., W.Z. and Y.L.; Software, Y.T., T.L. and Y.L.; Writing—original draft, Y.T.; Writing—review and editing, Y.Z. All authors have read and agreed to the published version of the manuscript.

**Funding:** This work was supported by the Key Research and Development Program (2019GHY112033), the National Natural Science Foundation of China (No.51609207) and the Foundation for Outstanding Young Scientist in Shandong Province (No.BS2014HZ021).

**Acknowledgments:** We are grateful for assistance with Yantai University Civil Engineering Laboratory, China.

**Conflicts of Interest:** The authors declare no conflicts of interest.

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
