# Peer review of "The Migration Law of Iron during the Process of Water Icing"

_water, doi:10.3390/w12020441_

Round 1
Reviewer 1 Report
Note that I am attaching also the pdf file with the same information, which might be easier to read
Manuscript ID: water-678358
“The Migration Law of Total Iron during the Process of Water Icing”
Report no 1 by Jakub Ciazela
Summary:
The manuscript by Yuanqing Tang et al. presents a concise experimental study on the partitioning of iron between liquid water and water ice. A series of experiments tests three key factors: 1) thickness of the ice (which could be treated rather as relative volume of the ice in my opinion), 2) temperature of icing, and 3) initial concentration of iron in the water. The study allows for better predicting iron concentrations in under-ice water and ice during, and after icing, which in my opinion may have important implications for biochemical processes. The conclusions are justified by the results. The manuscript is within the scope of the journal, complete, and concise with the train of authors’ thought easy to follow and the language generally clear. Still, I strongly suggest a native speaker to polish the text. I did not focus on copyediting tasks but gave many examples of where the language could be improved among the minor issues. I also find a number of areas where the scientific aspects of the paper can be improved, especially at the sections of methods (missing details) and conclusion (too modest). In addition, I miss more discussion and comparison with the results obtained by Sun et al. (2018), which is cited but it is not entirely clear (at least to me) how the results of Tang et al. are innovative with respect to Sun et al.. Providing these improvements along with many major and minor issues listed below can be achieved by the authors, I would recommend publication. Overall, I judge these improvements to amount to a major revision. I agree to read the revised manuscript again if the Editor finds it necessary.
Major issues:
From line 1 on:
Throughout the manuscript you sometimes use Total Iron and sometimes iron. The name ‘Total Iron’ is especially confusing in the title and the abstract. As you do not discuss iron speciation at all I would recommend to stick to iron throughout the manuscript. Then in the method section, you may mention that the method you use measures the total iron.
Lines 95-114:
Is the name of this paragraph appropriate? This is related more to analytical than experimental part. In addition, I cannot find any description of analytical technique used to determine the iron content (wet chemistry/AAS/OES/ICP-MS?). What was the name of the instrument (if any) and in which lab the measurements were performed. I miss also key information on the purity and providers of various reagents (ferrous ammonium sulfate, sulfuric acid, water).
Line 120:
The results of Sun et al. (2018) must be discussed to larger extent as he has investigated also iron among other heavy metals.
Lines 127-145 and Figure 3:
This text is very difficult to follow and in places would fit much better into the caption of Figure 3, where the caption is much too short. In addition, the figure axes need some quantification. I suppose that the eutectic point and shapes of the liquidus lines for iron-water system are known from the literature, and thus y (temperature) and x (composition) could be completed with real numbers. What does constant pressure mean – 1 bar? Why do you put (NH4)2Fe(SO4)2·6H2O (mg/L) below the axis if you (should) discuss pure water-iron system? Why do you put (s) after H2O and Iron? What is the meaning of T/K?
Figure 4:
This is a nice illustration of the process. Yet a scale bar would be useful to understand the nature of this process better. What is the source of data for this model – only literature, or you have some structural or chemical results made on your material that could demonstrate it? If so, provide the data, if not state your literature source in the caption. What do you mean by ‘capture iron’ – native iron or iron‐rich ice?
Section 3.2:
I rather disagree that this factor can be ice thickness as your liquid water reservoir is small, in some experiments even smaller than the produced ice body. Therefore, in my opinion, the iron concentration in the remaining water is controlled by volumetric proportions between liquid water and ice and not the ice thickness. In natural conditions, whether your ice thickness is 4 cm or 20 cm would have little impact on the iron content in under-ice water. What will more likely matter will be the volumetric relationship between the ice and water, and therefore I suppose your results will be especially relevant for small and shallow lakes or ponds.
Figures 5–10:
You should provide also a table summarizing all your results (at least as a supplementary material) as the figures themselves do not allow reading the exact concentrations of iron, especially in the ice. In addition, the meaning of error bars is not explained in the figure captions. The meaning of K should also be explained in the relevant captions.
Conclusions:
There should be more link with the rationale of this study that you describe in the introductions. One aspect completely not covered afterwards is ‘treatment of iron‐containing wastewater by using freezing concentration effect’. Yet more important would be potential implications of your results on natural ecosystems. For example, 1) which lakes in terms of depth, size, latitude and other factors would be most vulnerable for seasonal increase of Fe concentration in under-ice water, and 2) what ecosystems or animal and plant species would be most affected by the increased Fe concentrations in under-ice water.
Minor issues:
Abstract:
Line 11:
I am not sure if the formulation of ‘migrated from ice to under-ice water’ is correct. Please, try to reword, perhaps something like ‘partitioned stronger to under-ice water than ice’ would be more appropriate?
Line 14-15:
I would be more interested to read how you explained it than just a statement that you explained it.
Line 16:
‘people’s attention’ is too broad. Please better identify your target group. You should also extract something from the revised conclusions to this abstract and tell the readers in which cases the increased Fe concentration is relevant and why.
Introduction:
Line 23:
I would suggest to use ‘>’ signs, for example >150-day, >50 million. This is valid for the entire manuscript.
Line 24:
will reduce à reduces
Line 25–26:
The clause ‘the light penetration will be reduced which is insufficient for photosynthesis’ sounds awkward. Did you mean ‘reduced to a level insufficient for photosynthesis’ ?
Lines 28–29:
results in the decrease of (…) biodegradation à lowers dissolved oxygen concentration inhibiting photolysis and biodegradation
Line 32–33:
I do not think decreasing ice duration makes your topic more urgent — you could rather suppose … less urgent. I would remove this sentence from your argumentation.
Line 33–34:
This is an interesting fact but not an issue per se. The 2% may be still a big number in absolute terms and the remaining 98% likely covers equally important topics.
Lines 34–36:
This sentence does not bring much information and is badly written. I would suggest to remove it incorporating what is needed into the next sentence.
Lines 39–40:
Chemical, physical variables à chemical, physical properties.
Lines 39–40:
nitrogen, phosphorus, and dissolved organic carbon contents
Line 44–45:
please rewrite, either ‘cause cancer’ or ‘are cancerogenic’
Line 49–51:
This statement comes out of the blue. Firstly, the law of salt migration had not been introduced before. Secondly, the logical link between the first clause of this sentence starting with ‘if’ and the second clause of this sentence starting with ‘it will’ is completely unclear to me.
Lines 55–57:
Split in two sentences
Line 58–59:
This aspect never comes back in the manuscript but in my opinion it should, either in discussion or conclusions.
Materials and methods:
Line 66–67:
Give full names for height and diameter, calculate the volume and also report it here as you refer to it later.
Line 66–67:
Please, explain briefly the principle of this measurement. It is not clear to me in the current text.
Line 68–69:
Probably unnecessary detail as you needed only −25°C
Line 69:
Please, be more specific what you mean by temperature fluctuation – do you mean temperature deviation from the target temperature during experiments? For example, you targeted −5.0°C and registered temperatures between −4.5 and −5.5°C? Was that within 0.5°C for all the experiments or differed depending on the targeted temperature. What was inaccuracy of the thermometer itself?
Line 72:
T‐bracket might be confusing as T means also temperature. Please find another name, eventually T-shaped bracket would probably be fine.
Lines 83, 92:
The information in parenthesis was provided before. I think you do not need to repeat it three times.
Line 97:
Please describe briefly how you cut the ice. Discuss potential contamination issues if you used a steel tool.
Line 97:
I would avoid yielding artificial definitions. You can simply write 0–4 cm, 4–8 cm etc. intervals throughout the manuscript as you do not discuss it so much. It will be easier to follow by the readers.
Line 98:
revise to ‘melted in beaker at room temperature’
Lines 101–102:
How many triplicates were prepared (for all the samples or just selected?). What do you mean in this sentence – standard deviation of the triplicate was always < 5% of the mean of the three triplicates?
Line 104:
Please, improve the formatting of the chemical formula.
Line 104:
What does (1 + 1) mean?
Lines 109–114:
This is not pertinent neither to the experimental nor analytical part and can probably form a small subsection (2.4) itself.
Line 112:
Perhaps, you should provide a number for this equation.
Line 113:
I suggest to revise into ‘, where Ci is the average (…) and Cw is (…)’
Results and discussion:
Line 119:
Again, suggested migration from ice to under-ice water seems not to be correct
Line 120:
Remove ‘In view of this phenomenon,’
Lines 123–127
I am not sure if formulation ‘as follows’ is a good way to end a paragraph. Try to find a better way to link the two paragraphs.
Figure 2:
This figure does not contain anything in addition to Figures 6, 8, and 10. I would suggest to remove this figure.
Lines 151–153:
This sentence is difficult to read, perhaps due to incorrect syntax. Please reword.
Lines 153–155:
I struggle to understand this part ‘the ice embryo grows to a critical size and continues to grow by continuously increasing the water molecules’
Lines 162, 171, 177:
In these three lines and throughout the manuscript, I would suggest to introduce the figure references in parentheses (Fig. 4) instead longish descriptions such as ‘As shown in Fig. 4’
Line 176:
will form à forms/formed
Line 178:
I suggest: ‘(…) faster increasing the pore channel density and iron content’
Line 185:
What do you mean by the pollutants? Iron? If so, please, be precise and state it.
Figures 5, 7, 9:
As mentioned before I would suggest to replace Ice 1, 2 etc. with 0-4 cm, 4-8 cm etc.
Line 198–199:
You had explained this process in lines 180–181 using identical phrases. Please, try to condense you descriptions to avoid redundant repetitions.
Lines 198–202:
In this paragraph, when you refer to positions in the literature (48, 49, 50) you should rather write cf. 48, cf. 49, cf. 50 as you do not describe a general law but the processes of your experiment.
Line 199–202:
This sentence is very long and hard to follow. For example ‘which resulted in the volume of newly formed ice crystals became larger’ could be better ‘which resulted in the larger volume of newly formed ice crystals’, I guess.
Line 220:
Remove ‘As’
Line 223:
Again, if you mean your experiment, add cf. before the citation.
Lines 224-225:
Removal from where? What pollutants?
Line 235:
Refer to Fig. 9 in parentheses after “ice after icing”
Line 236:
Refer to figure 10 in parentheses after “initial concentration”
Line 243–244:
Please introduce definitions of viscosity and diffusion coefficients. They had not been discussed before. Also explain why they increased or decreased.
Lines 245–246
Again, you should use cf. before the references
Lines 248–249
Again, refer to the relevant figures
Conclusions:
Lines 252–253:
I feel the first sentence is not needed here.
Lines 253–254:
I think migrate is not good term here. In my opinion, it is better to write that iron partitions stronger to liquid water than ice.
Line 258:
Perhaps the meaning of K should be recalled in the conclusions so that accidental readers can understand the conclusions independently on the other parts of the article.
Line 260
I do not understand the term ‘are conductive to’. Do you mean favorable for?
Line 262–266
I feel this paragraph does not bring anything concluding to the paper. You can remove it, or alternatively try to move into discussion section
Authors contributions:
Lines 268–269:
Please, be more specific what you mean by the data curation, formal analysis, methodology and software
Acknowledgements:
Please be more specific, and also correct the grammar in the sentence
References:
Check formatting of positions 4., 15., 18–20., and most of 22–50.
Position 21 — I am not sure what the journal policy is — can an abstract be cited?
References used in this review:
Sun, C., Li, C., Liu, J., Shi, X., Zhao, S., Wu, Y., 2018. First-Principles Study on the Migration of Heavy Metal Ions in Ice-Water Medium from Ulansuhai Lake 1–13. doi:10.3390/w10091149

Author Response
Dear Reviewer
Thank you very much for your valuable suggestions. I have submitted a reply letter in word version.

Reviewer 2 Report
Dear Authors,
in the attachment I send comments to the article.
Best wishes,
EL

Author Response
Dear reviewer
Thank you very much for your valuable suggestions. I have submitted a reply letter in word version.

Round 2
Reviewer 1 Report
see the attached file

Author Response
I submitted a Word version of the reply letter, thank you very much for your advice and help.
